# Fabrication and Thermal Performance of a Polymer-Based Flexible Oscillating Heat Pipe via 3D Printing Technology

**DOI:** 10.3390/polym15020414

**Published:** 2023-01-12

**Authors:** Zhaoyang Han, Chao Chang

**Affiliations:** Institute of Marine Engineering and Thermal Science, Marine Engineering College, Dalian Maritime University, Dalian 116026, China

**Keywords:** 3D printing, flexible oscillating heat pipe, thermal resistance, thermal performance

## Abstract

As flexible electronic technologies rapidly developed with a requirement for multifunction, miniaturization, and high power density, effective thermal management has become an increasingly important issue. The oscillating heat pipe, as a promising technology, was used to dissipate high heat fluxes and had a wide range of applications. In this paper, we reported the fabrication and heat transfer performance evaluation of a polymer-based flexible oscillating heat pipe (FOHP) prepared using 3D printing technology. The 3D-printed inner surface presented excellent wettability to the working fluid, which was beneficial for the evaporation of the working fluid. Ethanol was selected as the working fluid, and the influence of the filling ratios range of 30–60% on heat transfer performance was analyzed. It was found that a 3D-printed FOHP with a filling ratio of 40% presented the best heat transfer performance with the lowest thermal resistance, and the fabricated heat pipes could be easily bent from 0° to 90°. With the best filling ratio, the thermal resistance of the FOHPs increased with larger bending angles. In addition, the 3D-printed FOHP was successfully applied for the thermal management of flexible printed circuits, and the results showed that the temperature of flexible printed circuits was kept within 72 °C, and its service life was guaranteed.

## 1. Introduction

With the rapid development of microelectronics technology, the development of flexible electronic devices and foldable electronic devices has become one of the essential trends in the future to meet the demand for portability and miniaturization [1,2,3]. For example, folding screen phones and wearable electronic devices have been widely applied in many important industrial fields in the past few decades. The compact structure and complex functions of this type of product led to increased heat generation, which seriously inhabited the service life of electronic devices [4,5,6]. Therefore, there was an urgent need for an advanced cooling device with excellent thermal conductivity to solve these problems mentioned above. Heat pipes, as common thermal management devices, were widely used in the field of electronic heat dissipation due to their simple structure and high reliability [7,8,9,10,11,12]. The heat pipe may be classified as a thermosyphon, conventional heat pipe, loop heat pipe (LHP), micro heat pipe, rotating heat pipe, vapor chamber, and oscillating heat pipe (OHP). Among them, the oscillating heat pipe, which functions via a thermally excited oscillating motion, was considered one of the most promising methods to solve the problem of compact heat dissipation based on its high thermal performance, simple construction, and small size [13,14,15]. Currently, OHPs have demonstrated quite promising potential for many applications in electronic cooling [16,17], aerospace [18], waste heat recovery [19], solar energy [20], and other important industrial fields [21,22,23].

Most current heat pipes, however, are composed of rigid metal materials, such as aluminum and copper. For the thermal management of flexible electronic systems, inflexible, rigid heat pipes could not be folded repeatedly and had high thermal contact resistance when fitted with flexible electronic devices. In facing issues of the heat dissipation of flexible electronic components, the application of rigid heat pipes was greatly limited. In contrast, flexible heat pipes may transfer heat at different bending angles and maintain high heat transfer efficiency under repeated bending experiments, which has now become a new trend in the field of electronic heat dissipation [24,25,26].

The flexible heat pipe usually consists of three sections: an evaporating section, a condensing section, and an adiabatic section. According to previous research, current flexible heat pipes may be divided into two categories: single articulated flexible heat pipes and non-articulated flexible heat pipes [24]. The single articulated flexible heat pipe uses a flexible connector as the adiabatic section to connect the evaporating section to the condensing section. The flexible connectors are usually composed of metal bellowed tubes [27,28], polyurethane tubes [29], and fluoro rubber tubes [30,31]. For example, Jaipurkar et al. [28] designed a flexible heat pipe that was composed of two copper tubes and a metallic bellow. Water was used as the working liquid, and the evaporating section was connected to the condensing section using a stainless steel bellow tube, which could be easily bent to a desired angle ranging from 5° to 20°. Yang et al. [29] reported a flexible heat pipe using a polyurethane tube as the connection material. It was found that the thermal performance of this flexible heat pipe was slightly affected by bending when the heat pipe was bent to an angle between 30° and 120°. Qu et al. [30] designed and fabricated a flexible oscillating heat pipe with a fluoro rubber tube connector. This heat pipe could be deformed at various structural styles, and the experimental results showed that the heat pipe had excellent flexibility and acceptable heat transfer capability. In fact, due to the characteristics of metal materials, metal connectors did not have good flexibility, which meant it was difficult to complete a large bending angle. Although the polymer connector was relatively flexible, the deformation of the polymer tube would damage the internal wick structure and deteriorate the heat transfer performance of the heat pipe.

Most of the non-articulated flexible heat pipes comprise polymer materials to accomplish bending directly by using shell materials. Compared to the single articulated flexible heat pipe, the non-articulated flexible heat pipe may be deformed to achieve a larger bending angle by increasing the bending radius, thus greatly protecting the internal structure of the heat pipe and improving flexibility and reliability [32,33,34,35]. A great amount of research was carried out on polymer-based flexible heat pipes. Christopher et al. [32] fabricated a flat flexible heat pipe using a commercial film as its shell, which consisted of low-density polyethylene, polyethylene terephthalate, and aluminum layers. The wick structure was chosen as a three-layered sintered copper mesh. When the heating power was 25 W, the thermal resistance of the polymer-based heat pipe was 23% of that of a copper block of the same size. Hsieh et al. [33] prepared a flexible heat pipe using silicone rubbers. This heat pipe had better heat transfer performance than the vertical working condition at a bending angle of 15°. In addition, it could work stably at a power of 12.67 W. Lim et al. [34] developed a polymer flexible pulsating heat pipe of which the shell consisted of multilayer laminated films and low-density polyethylene. To further improve the reliability of the heat pipe, it was sealed using a plus indium coating, which substantially increased the lifetime of the heat pipe.

Although a large amount of research on the optimized design and fabrication of polymer-based flexible heat pipes was performed, the conventional fabrication process was complex, high-cost, and inefficient. Therefore, it was necessary to develop a simple, efficient, and fast production method to fabricate flexible heat pipes. Fused deposition modeling (FDM) was one of the most widely used 3D printing technologies in manufacturing, production applications, and mechanical modeling. It uses a heated nozzle to melt and extrude thermoplastic polymer materials and deposits the extruded polymer filaments layer by layer, according to a designed pattern. Many thermoplastic polymer materials were applied for FDM printing, such as polylactic acid (PLA), acrylonitrile butadiene styrene (ABS), thermoplastic polyurethane elastomer (TPU), etc. Due to its fast production, low cost, and capability to create complex parts, FDM technology is widely used in the automotive industry, shipbuilding, aerospace, regenerative medicine, and other fields [36,37,38,39,40,41,42,43,44,45]. In this work, a 3D-printed flexible oscillating heat pipe (FOHP) with a size of 87 mm × 25 mm × 5 mm was designed and fabricated using thermoplastic polyurethane elastomer (TPU) as the raw material, according to the technical features of 3D printing rapid manufacturing. Ethanol was chosen as the working fluid, and the effects of different filling ratios, heating powers, and bending angles on the thermal resistance of the 3D-printed FOHP were experimentally investigated. Finally, the developed FOHP was applied in the thermal management of flexible printed circuits.

## 2. Materials and Methods

### 2.1. Materials

The thermoplastic polyurethane (TPU) filament with a diameter of 1.75 mm as the raw material for the 3D printer was ordered from Guangdong Ruiben Co., Ltd. (Guangzhou, China). A ceramic electric heater with a size of 25 mm × 25 mm was bought from Wujiang Zhongzheng Electric Technology Co., Ltd. (Tianchang, China). Ethanol was purchased from Shanghai Aladdin Reagent Co. Ltd. (Shanghai, China). Thermal conductive grease was ordered from Shanghai Taipu New Materials Technology Co., Ltd., (Shanghai, China). The flexible printed circuits were brought from Jiangsu Keyiman Electric Heating Technology Co., Ltd. (Yancheng, China). The thermal insulation material with a thermal conductivity of 0.05 W/m K was ordered from Luyang Energy-Saving Materials Co., Ltd. (Zibo, China).

### 2.2. Fabrication Process of the Flexible Oscillating Heat Pipe

An oscillating heat pipe with an internal serpentine flow channel structure used the working fluid to transfer heat by oscillating back and forward between the evaporating section and condensing section. The diameter of the inner channel was an important parameter for the operation of the oscillating heat pipe, and its maximum value was determined using the following formula [46]:(1)$$D\leq 1.84\sqrt{\frac{\sigma }{g\left(\rho_{l}-\rho_{v}\right)}}$$
where *σ* is the surface tension, g is the acceleration of gravity, $\rho_{l}$ is the density of the liquid, and $\rho_{v}$ is the density of the vapor.

In this work, we adopted a fused deposition modeling (FDM) technique to prepare the FOHP, which is one of the most widely used 3D printing technologies due to its low cost, fast production, and easy operation. As shown in Figure 1a, the filament was heated to a semi-liquid state using a 3D printer’s extruder and then extruded through a layer-by-layer process according to a specific pattern. The nozzle was maneuvered using a program that allowed the flow of the melted 3D-printed material to be switched on or off. After the melted material solidified, the printed object was obtained. Figure 1b presents the designed structure of the FOHP in this study, of which the shell size was 87 mm × 25 mm × 5 mm, and the turn number was 4. Ethanol was used as the working fluid, and the diameter of the inner channel was set as 2 mm, which satisfied Equation (1).

To achieve the bending capability of the fabricated oscillating heat pipe, among the common 3D-printed polymer materials, a TPU with a low density and good flexibility was chosen as the raw material. The FOHP was created using an FDM 3D printing system (Appendix A and Note S1), and the temperature of the nozzle, temperature of the platform, and scanning speed were set to 210 °C, 60 °C, and 30 mm/s, respectively. The nozzle, which was controlled using the system program, melted the TPU filament according to the structural information of the oscillating heat pipe and deposited it on the surface of the preheated platform layer by layer until the entire heat pipe was produced.

According to our previous studies [12,22,42], we adopted a filling-back method to fill the working fluid into the 3D-printed FOHP. The FOHP was first evacuated to a vacuum below 5 Pa, and then the working fluid was charged into the entire FOHP by the pressure difference. We drew out a certain amount of the working fluid and sealed the FOHP. The filling ratio was generally defined as the ratio between the remaining working fluid and the whole inner space of the FOHP. During the experiment, the effects of the filling ratios (30%, 40%, 50%, and 60%), heating powers (1 W~8 W), and bending angles (0°, 45°, and 90°) on the thermal performance of the FOHP were investigated.

### 2.3. Experimental Setup

The experimental setup for evaluating the heat transfer performance of the FOHP is displayed in Figure 2. The measurement apparatus mainly included a ceramic electric heater, DC power supply, cooling block, cooling bath, data acquisition system, and computer. The heating unit consisted of a ceramic electric heater with a size of 25 mm × 25 mm and an adjustable DC power supply. The heating power input added to the FOHP was controlled by adjusting the voltage and current of the DC power supply. The condensing section of the FOHP was cooled using a cooling block, which was connected to a cooling bath, and the cooling temperature for this experiment was set at 20 °C. Six thermocouples (T1~T6) were used to monitor and record the real-time temperature variation of the FOHP. Three thermocouples were uniformly arranged in the center of the evaporating section and condensing section of the FOHP, respectively. The measured temperature data were collected using a data acquisition system and then transmitted to the computer. In addition, to improve the accuracy of the measurement data, conductive silicone grease was applied between the contact surfaces to reduce the contact thermal resistance. During the experiments, the entire FOHP was wrapped in thermal insulation materials with a thickness of more than 5 cm to minimize heat loss. By comparing the inlet and outlet water temperatures, the maximum heat loss of the system was 1.43%.

Thermal resistance (*R*) as an important indicator was used to evaluate the heat transfer performance of the heat pipe. The thermal resistance of the 3D-printed FOHP was obtained using the following equation:(2)$$T=\frac{1}{N}\sum_{i=1}^{N}T$$
(3)$$R=\frac{\bar{T_{e}}-\bar{T_{c}}}{q}$$
where N, q, $\bar{T_{e}}$, and $\bar{T_{c}}$ are the number of thermocouples in the evaporating section or condensing section, the heating power input, and the average temperature of the evaporator section and the condenser sections, respectively.

### 2.4. Characterizations

The FOHP was fabricated using an FDM 3D printer (Creality 3D Ender-3) with an accuracy of ±0.1 mm. The microstructures of the inner surface of the FOHP were observed using a field emission scanning electron microscope (SEM, Sirion 2000, FEI). The contact angle was characterized with a contact angle measuring device (FM40Mk2 EasyDrop, KRUSS GmbH). The temperature variation of the 3D-printed FOHP was monitored and recorded using six K-type thermocouples (Omega SMPW-TT-K, resolution ~ 0.1 **°**C), which were connected to a multichannel data acquisition system (Agilent 34972A, Agilent Technologies Inc., Beijing, China). To analyze the thermal performance of the heat pipe, a cold block was connected to the FOHP at the condensing section, and a cooling bath (Julabo Bilon Equipment, Seelbach, Germany) offered circulated cooling water at a consistent temperature of 20 °C. An infrared camera (HM-TPH36-10VF/W HIKVISION, Hangzhou, China) was used to monitor the surface temperature of the FOHP.

## 3. Results and Discussion

Figure 3a displays an optical image of the prepared FOHP, which was fabricated using the 3D printing technology mentioned above. The fabricated FOHP was composed of thermoplastic polyurethane (TPU) with a size of 87 mm (length) × 25 mm (width) × 5 mm (height), and the error was ±0.1 mm. Figure 3b presents the fabricated FOHP with excellent flexibility, easily bent to a bending angle of 90°.

To characterize the surface morphology and wetting performance of the inner channel in the 3D-printed FOHP, the prepared FOHP was cut into two halves in the axial direction. Figure 4a exhibits the inner surface morphology of the flow channels in the FOHP, which was observed using a scanning electron microscope (SEM). As shown, the inner surface of the channel presented small groove structures, which were attributed to the features of the FDM manufacturing process. The unique groove structure would improve the roughness of the inner surface and further facilitate the evaporation of the working fluid. The internal channels of the FOHP consisted of four turns. The cross-section of the grooves was circular, and the diameter was 2 mm. The distance between the two adjacent internal channels was 1 mm.

In addition, we also measured the contact angle of the inner surface, as shown in Figure 4b,c. When an ethanol droplet dropped on the inner surface of the FOHP, the measured contact angle was 13°, indicating that the inner surface presented excellent wettability to the working fluid. Therefore, the groove structure and wettability of the inner surface not only provided a strong capillary capability but also facilitated the evaporation of the working fluid.

Figure 5 shows the thermal performance of the 3D-printed FOHP with different filling ratios when the heating power input increased from 1 W to 8 W. The quantity of working fluid played an important role in the thermal performance of the FOHP. In general, when too much working fluid was filled into the FOHP, the liquid plugs would restrict the oscillating motion, leading to poor thermal performance. On the contrary, when the FOHP was loaded with too little working fluid, there would be insufficient liquid to absorb and convert the thermal energy added onto the evaporating section into the kinetic energy of the working fluid, thereby resulting in the phenomenon of dry-out. Therefore, an optimum filling ratio should exist in the oscillating heat pipe. Figure 5a shows the temperature difference evolution of the FOHP filled with different filling ratios (30%, 40%, 50%, and 60%) at various heating power inputs. As shown, when the heating power input increased from 1 W to 7 W, the FOHP with a filling ratio of 40% had the smallest temperature difference between the evaporating section and condensing section compared to the other three FOHPs. When the heating power input was 7 W, the temperature difference of the FOHP with a filling ratio of 40% was only 8.1 °C. However, when the heating power input was continuously increased to 8 W, the temperature difference of the FOHP increased sharply, indicating that the operating limitation was reached.

We also calculated the thermal resistance *R* according to Equations (2) and (3). As shown in Figure 5b, the thermal resistance of all the FOHPs decreased as the heating power input increased until the operating limitation appeared. Among them, the FOHP with a filling ratio of 40% presented the optimum thermal performance with the lowest thermal resistance compared to the other three heat pipes with a filling ratio of 40%, 50%, and 70%. When the heating power input was 7 W, the thermal resistance of the FOHP with a 40% filling ratio was only 1.16 °C/W. Based on the law of Fourier, the effective thermal conductivity of the FOHP, k, may be obtained using the following formula:(4)$$q=kA\frac{\Delta T}{l}$$
where *A*, l, and ΔT are the cross-section area of the FOHP, the distance between the evaporating section and the condensing section, and the temperature difference between the evaporating section and the condensing section, respectively. According to Equation (4), the maximum effective thermal conductivity of the 3D-printed FOHP was 514 W/m K when the heating power input was 7 W. As a consequence, the optimum filling ratio for this 3D-printed FOHP was 40%, and the FOHP with a filling ratio of 40% would be applied in further experiments.

In the following study, the FOHP filled with 40% working fluid was used to investigate the effect of bending on the heat transfer performance. Figure 6 exhibits the thermal resistance evolution of the 3D-printed FOHP tested at different bending angles. As shown in Figure 6b, it should be noted that the thermal resistance of the FOHP decreases when the heating power inputs increases and a larger bending angle results in more significant thermal resistances, especially when a low heating power input is added to the evaporating section. As shown, without bending, the thermal resistance of the FOHP dropped from 27.5 °C/W at 1 W to 7.3 °C/W at 3 W and 2.8 °C/W at 5 W. When the FOHP was bent to 90°, the corresponding thermal resistances were 35.1 °C/W, 10.6 °C/W, and 4.4 °C/W. On the other hand, when the FOHP was not bent, the operating limit of the FOHP was 7 W. While the FOHP was bent to an angle of 90°, the operating limit of the FOHP dropped to 5 W. Therefore, the experimental results showed that bending would cause additional thermal resistance of the FOHP and decrease the operating limitation, but the additional thermal resistance was able to reduce by increasing the heating power input at the evaporating section. In addition, it was clearly seen that when the heating power input was increased from 1 W to 5 W, this 3D-printed FOHP presented good thermal performance and low thermal resistance with bending angles ranging from 0° to 90°.

Finally, due to its excellent heat transfer performance and flexibility, we further demonstrated our developed FOHP for the thermal management of flexible printed circuits. As shown in Figure 7a, a flexible printed circuit, which could generate heat ranging from 2 W to 6 W, was attached to the evaporating section of our FOHP with thermal grease. A cooling block with a constant temperature of 20 °C was placed on the condensing section of the FOHP. In addition, a K-type thermocouple, which was placed on the surface of the flexible printed circuit, was used to monitor and record its real-time temperature evolution. The heating generated by the flexible printed circuit was first transferred to the evaporating section of our developed FOHP, exciting oscillating motion in the pipe, and then carried to the condensing section, where the heat was further dissipated by the cooling water.

Figure 7b displays the temperature evolution profiles of the flexible printed circuit surface over time. As shown, the surface temperature would quickly reach more than 100 °C in less than 1 min when the flexible printed circuit was powered without the operation of our FOHP, thereby resulting in the damage of the circuit. In contrast, when the flexible printed circuit was connected to the flat-plate OHP, the surface temperature would quickly reach a steady working state at 52 °C at a power of 2 W, thereby prolonging the service life of the circuit. When the power was increased to 4 W and 6 W, the surface temperature of the FOHP was only 60 °C and 67 °C, respectively. In addition, the influence of bending angle on the heat transfer performance of the FOHP was also investigated. When the FOHP was bent to 90°, the surface temperature of the circuit rose to 72 °C at a power of 6 W, which was slightly higher than that in the no-bending situation.

To further evaluate the thermal management performance of the FOHP, when the flexible printed circuit reached a steady state, thermal infrared images at different heating power inputs were collected, as shown in Figure 7c. It was obvious that a slight temperature gradient existed in the FOHP, which indicated that the FOHP had uniform heat transfer. More important, the FOHP could be bent as the flexible printed circuit was bent. Even though the FOHP was bent, it still presented excellent thermal performance and uniform heat transfer. Therefore, it was highly expected that the fabricated 3D-printed FOHP would be able to provide a new approach for researchers to solve the problem of the thermal management of flexible electronic devices.

## 4. Conclusions

In summary, we successfully developed a flexible polymer-based oscillating heat pipe using FDM 3D printing technology, which completed the fabrication of a FOHP with complex internal channels created in a short production cycle. Ethanol was used as the working fluid for the FOHP, and the effect of the filling ratios, heating power inputs, and bending angles on the thermal performance was systematically analyzed. The results showed that the FOHP with a filling ratio of 40% had the lowest thermal resistance and largest operating limitation, and the maximum effective thermal conductivity was 514 W/m K at a power of 7 W. In addition, the fabricated 3D-printed FOHP demonstrated excellent thermal performance with low thermal resistances within a range of bending angles from 0° to 90°. At last, the 3D-printed FOHP was integrated with flexible printed circuits to verify its thermal management performance. It not only maintained the normal operation of flexible printed circuits but also prolonged their service life. Besides high-power flexible printed circuits, our developed 3D-printed FOHP is expected to be applied in high-power flexible LEDs, wearable electronics, flexible multifunctional sensors, and other flexible systems which involve the requirement of heat dissipation within a compact space.

## Figures and Tables

**Figure 1 polymers-15-00414-f001:**
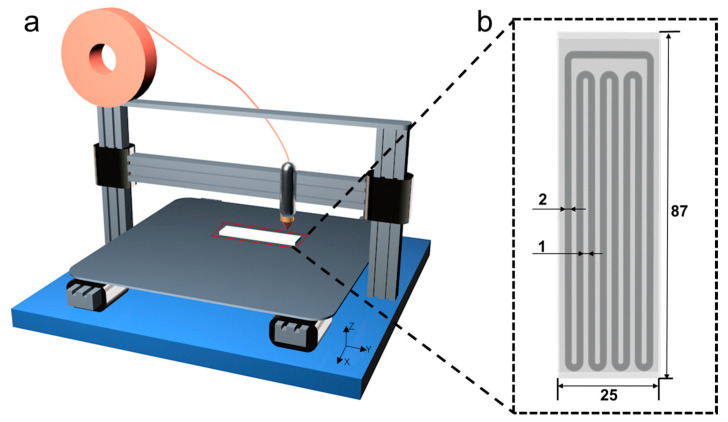
Schematic diagram of (**a**) the 3D printing process and (**b**) the structure of the 3D-printed FOHP.

**Figure 2 polymers-15-00414-f002:**
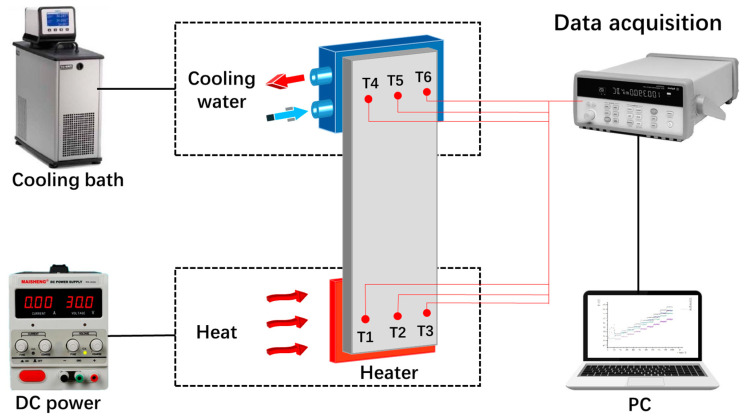
Schematic diagram of the experimental setup for thermal performance of the FOHP.

**Figure 3 polymers-15-00414-f003:**
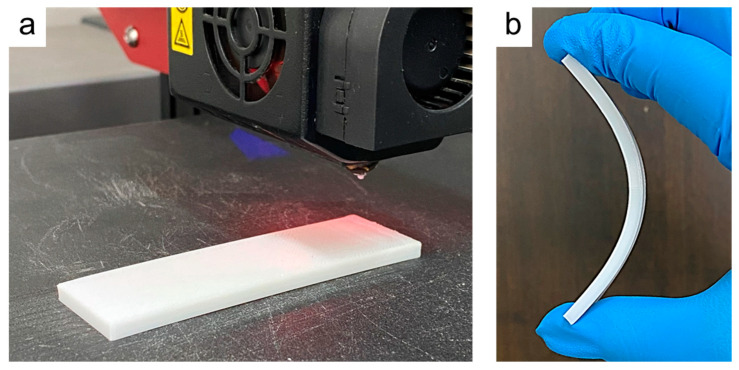
Optical images of (**a**) the 3D-printed FOHP and (**b**) the FOHP with a bending angle of 90°.

**Figure 4 polymers-15-00414-f004:**
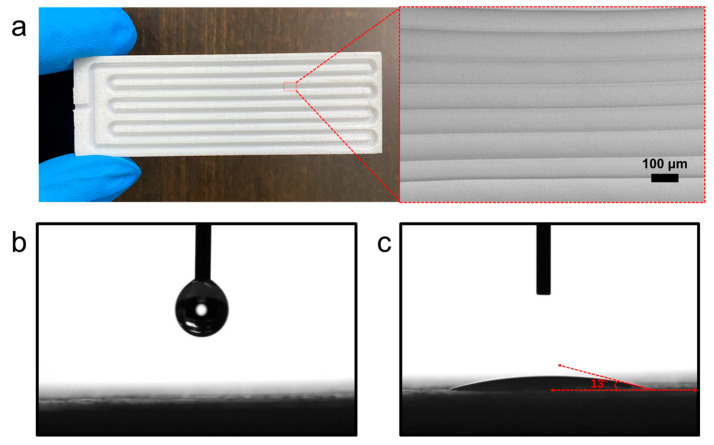
Photograph of (**a**) the internal serpentine flow channel structure of the FOHP and an ethanol droplet dropped (**b**) before and (**c**) after the inner surface.

**Figure 5 polymers-15-00414-f005:**
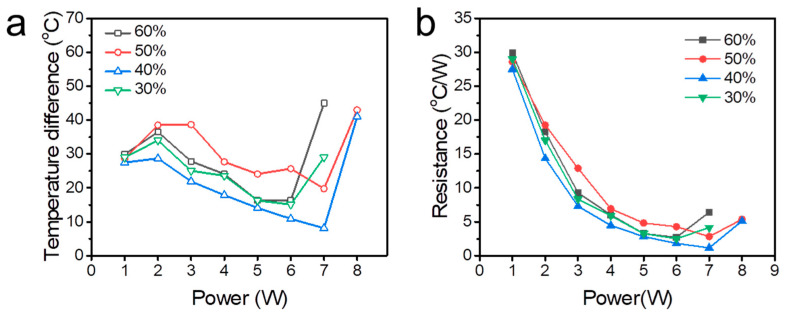
Thermal performance of the FOHP. (**a**) Temperature difference and (**b**) thermal resistance evolution with different heating power inputs.

**Figure 6 polymers-15-00414-f006:**
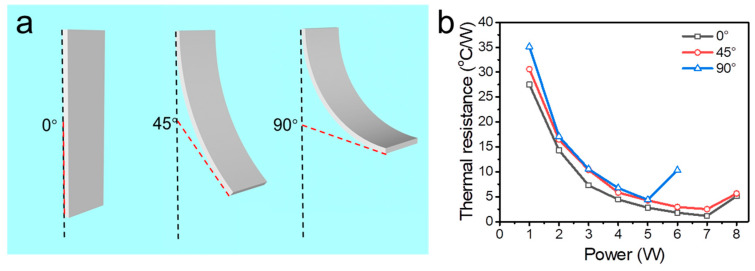
Thermal performance of the FOHP at different bending angles. (**a**) Schematic of the bent FOHP. (**b**) Thermal resistance evolution at different bending angles.

**Figure 7 polymers-15-00414-f007:**
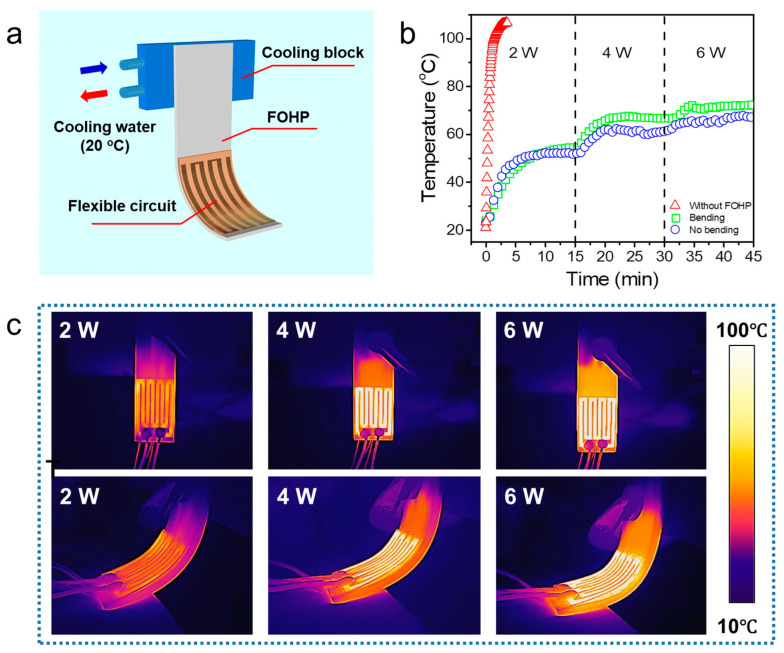
Thermal management of a flexible printed circuit by the 3D-printed FOHP. (**a**) Schematic of thermal management of a flexible printed circuit by a FOHP. (**b**) Surface temperature variation of the flexible printed circuit managed by the FOHP over time. (**c**) IR images showing different thermal performances by the FOHP at different heating power inputs.

## Data Availability

The data presented in this study are available upon request from the corresponding author.

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
