# Peer review of "Fabrication and Thermal Performance of a Polymer-Based Flexible Oscillating Heat Pipe via 3D Printing Technology"

_polymers, 2023, doi:10.3390/polym15020414_

Round 1
Reviewer 1 Report
see attached pdf

Author Response
The manuscript attempts to present a case about the fabrication and thermal performance of a polymer-based flexible oscillating heat pipe via 3D printing technology. The paper is interesting in its experimental part but the introduction about the 3D Printing technology is very weak. The list of references should definitely be expanded, since they are few in quantity.
My points are analytically listed below
Points for consideration:
Point 1 The introduction is very weak in terms of not describing 3D Printing technology. Insert a relevant paragraph in the end of the introduction and use the following references about FDM 3D Printers
ï‚· 10.1016j.jengtecman.2015.09.003
ï‚· 10.1016j.bushor.2011.11.003
ï‚· 10.5923j.mechanics.20211001.02
ï‚· 10.1016j.matpr.2021.09.074
ï‚· 10.1108JMTM-12-2016-0182
Reply: Thanks for the comments. In the revised manuscript, we added more description on 3D printing technology in the introduction section, and cited the following references on page 2 and 3.
“Fused deposition modeling (FDM) was one of the most widely used 3D printing technologies in manufacturing, production applications, and mechanical modeling. It used a heated nozzle to melt and extrude thermoplastic polymer materials, and deposited the extruded polymer filaments layer-by-layer according to a designed pattern. Many thermoplastic polymer materials were applied for FDM printing, such as polylactic acid (PLA), acrylonitrile butadiene styrene (ABS), thermoplastic polyurethane elastomer (TPU), etc. Due to its fast production, low cost and capability to create complex parts, FDM technology had been widely used in automotive industry, shipbuilding, aerospace, and other fields [36-41].”
Point 2 In line 106, change the word” wire” with the word “filament”.
Reply: We are very sorry for the mistake. In the revised manuscript, we changed the word “wire” with the word “filament”.
Point 3 In lines 136-143 mention the exact 3D Printer used and include a photograph apart from the one in figure 3a.
Reply: Thanks for the comments. The 3D printer used in the experiments was Creality 3D Ender-3, which was ordered from Shenzhen Creality 3D Technology Co., Ltd. The accuracy of 3D printer is ±0.1 mm. In the revised manuscript, we added the following description on the 3D Printer used and a photograph in the supplementary materials.
“The 3D printer with an accuracy of ±0.1 mm was used to fabricate the FOHP, and its build volume was 220 mm x 220 mm x 250 mm. The diameter of nozzle, maximum hot bed temperature, and maximum printing speed were 0.4 mm, 110 oC, and 200 mm/s, respectively. The layer thickness was 0.1~0.4mm.”
Point 4 In line 145-149 please include the number and type of elements used in the simulation.
Reply: Thanks for the comments. In this paper, we focused on the thermal performance of the 3D printed flexible heat pipe. The effects of filling ratios (30%, 40%, 50%, and 60%), heating powers (1 W~8 W), and bending angles (0o, 45o, and 90o) on the thermal performance of the flexible oscillating heat pipe were investigated. In the revised manuscript, we added the following description on the elements used in the work on page 4.
“During the experiment, the effects of filling ratios (30%, 40%, 50%, and 60%), heating powers (1 W~8 W), and bending angles (0o, 45o, and 90o) on the thermal performance of the FOHP were investigated.”
Point 5 In references, include the Doi addresses (in a link) of each reference, as this helps with reference matching.
Reply: Thanks for the comments. Following the reviewer’s suggestions, in the revised manuscript, we added the Doi addresses of each reference.

Reviewer 2 Report
The development and fabrication of the flexible polymer-based oscillating heat pipe presented in this article using FDM 3D printing technology could provide a new impulse for the development of flexible heat dissipation structures. The tests carried out have demonstrated the good properties of the material and the structure, providing results directly applicable to practical applications. The tests have been carried out and presented in sufficient depth. The organization and structure of the article is neat, logical and easy to follow; the published results are presented and analyzed objectively; the figures are well developed and informative. The article language is understandable, traceable, free of errors, very excellent work.
My suggestions for improving the quality of the article are only as follows:
Line 99-103: I suggest to delete from the introduction the following: "As a result, the FOHP with a filling ratio of 40% had the best thermal performance with the lowest thermal resistance."
"... and the experimental results demonstrated that the FOHP was able to maintain the normal operation of flexible printed circuits and prolonged the serving lifetime." There are two reasons for this: in the introduction we do not describe the results, but the objectives. These results will be repeated later.
line 189: the height is given as size 4mm, previously it was 5mm.
Author Response
The development and fabrication of the flexible polymer-based oscillating heat pipe presented in this article using FDM 3D printing technology could provide a new impulse for the development of flexible heat dissipation structures. The tests carried out have demonstrated the good properties of the material and the structure, providing results directly applicable to practical applications. The tests have been carried out and presented in sufficient depth. The organization and structure of the article is neat, logical and easy to follow; the published results are presented and analyzed objectively; the figures are well developed and informative. The article language is understandable, traceable, free of errors, very excellent work.
My suggestions for improving the quality of the article are only as follows:
Line 99-103: I suggest to delete from the introduction the following: "As a result, the FOHP with a filling ratio of 40% had the best thermal performance with the lowest thermal resistance."
"... and the experimental results demonstrated that the FOHP was able to maintain the normal operation of flexible printed circuits and prolonged the serving lifetime." There are two reasons for this: in the introduction we do not describe the results, but the objectives. These results will be repeated later.
Reply: Thanks for the comments. Following the reviewer’s suggestions, in the revised manuscript, we deleted the results in the introduction.
line 189: the height is given as size 4mm, previously it was 5mm
Reply: Sorry for the mistake. The height of the flexible oscillating heat pipe was 5 mm. In the revised manuscript, we revised this data.

Reviewer 3 Report
Please see attachment for my comments.

Author Response
In this paper, a polymer-based flexible oscillating heat pipe is fabricated using 3D printing technology and its thermal performance is measured experimentally. The experimental results such as temperature distributions, thermal resistances versus input power for different bending angles are reported.
Review results:
The manuscript should be published after implementing the following minor comments properly.
1- Address insulation material and estimation of heat loss in the text.
Reply: Thanks for the comments. To minimize the heat loss, we wrapped the whole FOHP by a porous thermal insulation material (Luyang Energy-Saving Materials Co., Ltd) with a thickness of more than 5 cm. The thermal conductivity of thermal insulation materials was 0.05 W/m K. The heat loss of the system was less than 1.43% (see the response to question 2). In the revised manuscript, we added the following description on the insulation material on page 3.
“The thermal insulation material with a thermal conductivity of 0.05 W/m K was ordered from Luyang Energy-Saving Materials Co., Ltd.”
2- Address flow rate of cooling water and its specific heat capacity. For the steady-state conditions, you can simply perform energy balance and estimate heat loss.
Reply: Thanks for the comments. The flow rate of cooling water was measured to be 42.68 ml/s. When the heating power input was 8 W, the maximum heat loss was 1.43% according to the equation. In the revised manuscript, we added the following description on the heat loss on page 4.
“Through comparing the inlet and outlet water temperature, the maximum heat loss of the system was 1.43%.”
3- Address accuracy of heat pipe dimensions, specifically for grooves in plus-minus.
Reply: Thanks for the comments. The accuracy of 3D printer is ±0.1 mm. We also measured the dimensions of the heat pipe and the grooves by a vernier caliper. The size of flexible oscillating heat pipe was 87 mm x 25 mm x 5 mm, and the diameter of grooves was 2 mm. The error was ±0.1 mm. In the revised manuscript, we added the printing accuracy on page 5.
4- There is a scale for heat pipe flow channel in Fig.4. But how depth are the grooves?
Reply: Thanks for the comments. The internal channels of the heat pipe consisted of 4 turns. The cross section of grooves was circular, and the diameter was 2 mm. The distance between the two adjacent internal channels was 1 mm. In the revised manuscript, we added the more description on the scale for the heat pipe on page 6.
“The internal channels of the FOHP consisted of 4 turns. The cross section of grooves was circular, and the diameter was 2 mm. The distance between the two adjacent internal channels was 1 mm.”
5- What camera did you use for surface temperature measurements in Fig. 7c?
Reply: Thanks for the comments. The surface temperature was measured by an infrared camera (HM-TPH36-10VF/W HIKVISION, Hangzhou, China). In our previous work, we also used the same camera to monitor surface temperature (10.3390/pr10050897). In the revised manuscript, we added the description on the camera on page 5.
“An infrared camera (HM-TPH36-10VF/W HIKVISION, Hangzhou, China) was used to monitor the surface temperature of the FOHP.”
6- It looks to me there is no transition from using ceramic heater to using flexible heater. Please explain more about why you changed heaters and what are their applications.
Reply: Thanks for the comments. The ceramic heater was able to provide a heating power of 0 W ~ 10 W, while the maximum heating power provided by the flexible heater was only 6 W. To investigate the thermal performance of the heat pipe, especially the operation limitation, we choose the ceramic heater. In addition, the ceramic heater was used to ensure that the evaporator section was not bent, and the effect of bending of the adiabatic section on the thermal performance of the FOHP was further investigated. In the application test, the flexible heater simulated the heat generation of flexible electronics such as flexible screens and wearable electronic devices, which was more similar to the operating conditions of FOHP in practical application.

Round 2
Reviewer 1 Report
I am pleased with the revisions that authors have made. In line 93-102 of the revised manuscript, authors have added a small paragraph regarding FDM 3D printing technology and relevant materials. Authors should also mention regenerative medicine as another field of application in line 101. Please add the following references after number 41.
· 10.3390/ijms232314621
· 10.3390/polym12122806
· 10.3844/ajeassp.2022.255.263
· 10.3389/fbioe.2020.586406
In this way, the references number will reach 45, which is marginally ok for such a manuscript.
After this small revision, I will consent for the article's publication.
Author Response
Reply: Thanks for the comments. Following the reviewer’s suggestions, in the revised manuscript, we added regenerative medicine as another field of application in line 101, and added the four references after number 41.
Thanks again for your great help!